# GLUECODE: A BENCHMARK FOR SOURCE CODE MACHINE LEARNING MODELS

## ABSTRACT

A multitude of machine learning models for source code have been proposed in the recent years capturing various aspects of the inherent rich structure and semantics of code. However, these models are commonly designed to perform well on a single task, failing to capture code's multifaceted nature. To address this, we present GLUECode, Global and Local Understanding Evaluation of Code, a benchmark of diverse tasks to evaluate machine learning models of source code.

Crucially, GLUECode accounts for the distinct characteristics of source code: (1) source code is highly structured and (2) source code is often composed of multiple interacting entities. Existing tasks incentivize researchers to create models and code representations that perform well on a single task - commonly focusing on local reasoning. GLUECode aims to allow researchers to experiment with multiple local and global source code representations, and evaluate these models on their ability to capture the diverse characteristics of source code, thus driving the community towards building robust source code models incorporating global reasoning. We present results for several baselines. The GLUECode tasks are challenging for the evaluated baselines; no model achieves convincing performance across all tasks. This indicates that there is ample room for progress on GLUECode.

## 1 INTRODUCTION

In recent years, there has been considerable interest in researching machine learning models on source code artifacts. Machine learning models have been used to address a variety of software engineering tasks, as the inherent rich structure of code has allowed machine learning researchers to explore new models and ideas. However, research has focused on single-purpose application models, targeting a single task each time while using varying source code representations and datasets. This impedes progress towards general-purpose machine learning models of code that can learn and reason across many tasks.

In this work, we present GLUECode (Global and Local Understanding Evaluation of Code), with the goal of measuring progress in source code modelling across a range of tasks that account for the diverse characteristics of software and require diverse reasoning capabilities over several thousands of software projects. As GLUE (Wang et al., 2018) and SuperGLUE (Wang et al., 2019) does for natural language, GLUECode highlights important aspects of reasoning about code: (1) since code in software is composed of multiple interacting entities, it includes tasks that leverage both *local* (single method) and *global* (multiple inter-related methods, information beyond the local method) reasoning to varying degrees. This is in contrast to most tasks and models that have been introduced so far that focus on local reasoning; (2) since source code mixes structured and unstructured information, GLUECode tasks leverage both kinds of information, and (3) since the space of modelling choices is large, we provide several source code representations ranging from raw text to abstract syntax trees (AST) and graph representations, lowering the barrier to entry and ease of experimentation.

The design space for source code models is extremely large and spans a wide range of source code representations. These range from the simplest (software metrics and $n$-grams), to very complex that fully take advantage of the structure and semantics of source code (such as graph-based representations). Even seemingly simple choices, such as how to preprocess identifiers, can be handled in many different ways and have disproportionate impact (Karampatsis et al., 2020). GLUECode aims to provide a unified benchmark to explore this design space.

We provide performance results on a set of baselines, ranging from simple neural architectures such as LSTMs and CNNs, to variants of pre-trained transformers. These models leverage purely local reasoning and limited amounts of structural information. We show that existing models perform well in a few tasks but fail to yield good results in others: In contrast to NLP, where (pre-trained) transformers outperform other models, we find that no single model of code consistently outperforms the others in all tasks.

Finally, while models can be evaluated on any single task in the benchmark in isolation (as the field is presently doing), a long-term goal of GLUECode is the creation of unified multi-task source code models that perform well across multiple tasks. A source code model that is jointly trained and can perform well on all the task in the benchmark would be a significant step towards more versatile models, that can, beyond the tasks they were trained, also adapt to downstream tasks, especially when there is not enough data. Given the performance of our baselines in the single-task scenario, defining a model that performs well across the board is thus very much an open problem.

## 2 THE GLUECODE BENCHMARK

Benchmarks are a common practice in machine learning and NLP, prominently featuring GLUE and SuperGLUE (Wang et al., 2018; 2019) among others. In the domain of machine learning on source code, several benchmarks have been proposed. However, in contrast to GLUECode, they consider relatively local contexts and do *not* incentivize non-local reasoning: Idbench looks at identifiers, (Wainakh et al., 2019), BigCloneBench (Svajlenko & Roy, 2015) and OJClone (Mou et al., 2016) at clone detection, and CodeSearchNet at a function-level text-to-code search (Husain et al., 2020). Finally, COSET concerns classifying small programs by their functionality in 38 classes (Wang & Christodorescu, 2019), and CoNaLa is a line-level text-to-code generation benchmark (Yin et al., 2018). In this section, we provide an overview of GLUECode. We first describe the software-specific characteristics that impact the choice of tasks, before detailing the dataset and the tasks involved. Details about other related benchmarks can be found in the Appendix D.

### 2.1 LOCAL VERSUS GLOBAL CONTEXT

Most existing machine learning models of source code work at the level of a single function or method. We call these *local models*, as they reason over the local context of a single software entity. This is in contrast to *global models* that reason over multiple software entities and scales. Global models are highly desirable since software systems are composed of multiple entities such as modules and functions, that communicate with each other. This composition of communicating entities dictates the behavior of a software system. For instance, a function may have a radically different behavior, depending on its arguments. Indeed, small local changes can manifest in large changes in behaviour in distant program locations. Only global models will be able to detect that. To push forward the state of the art, it is thus critical to focus on global models.

Fully global models are currently out of reach but GLUECode incentivizes building models that feature some form of global reasoning, in addition to local reasoning. Existing work uses simplified projections of global representations: the GNN works of Allamanis et al. (2017; 2020) look solely at file-level tokens, syntax, data and control flow information. CocoGum (Wang et al., 2020) uses class context represented as abstracted UML diagrams. LambdaNet extracts type dependencies in JavaScript into a single graph (Wei et al., 2020) for a few mid-sized projects (500-10k lines of code), ignoring syntactic information, code comments, etc. Finally, Func2Vec (DeFreez et al., 2018) computes function embeddings over an interprocedural call graph, ignoring local syntax, function arguments, etc. An extended related work discussion can be found in Appendix D.

Instead to reason over global contexts two limitations need to be overcome: First, time-consuming interprocedural static analyses need to be performed at scale. These require compiling projects and resolving all its dependencies. In GLUECode, we take a step towards this direction, by using the largest publicly available corpus of compilable Java code (Sec. 2.3). (2) Existing methods do not operate well on large and sparse inputs and thus representations are tailored to use only the necessary information. In GLUECode, we provide access to a variety of representations and propose a set of tasks that *cannot* focus solely on local or global information (Sec 2.2).

## 2.2 FLEXIBILITY IN REPRESENTATIONS OF CODE

Representations of source code in machine learning are a central topic of research. Source code has a known rich structure, as it can be unambiguously parsed, while valuable information is present in identifiers, literals, and comments, which are unstructured. As a result, there has been sustained work in exploring architectures and representations that leverage the different structural aspects of software, ranging from treating software as a textual artifact, to tree and graph-based representations. These representations come with distinct trade-offs.

Sequence-level models treating source code as text are simpler and easy to scale to large amounts of data, at the expense of obscuring the information obtained from distinct structural inter-relations in code. LSTM (Zaremba & Sutskever, 2014), CNN (Allamanis et al., 2016) and transformer (Husain et al., 2020; Kanade et al., 2020; Feng et al., 2020) based models for source code have been explored. Meanwhile, more structured models commonly learn from less data thanks to the provided structure, but are harder to scale as they require extensive pre-processing. Such models use a program's abstract syntax tree (AST) in Tree-LSTMs (Wei & Li, 2017), tree-based CNNs (Mou et al., 2014), or use linearized forms fed to sequence models (LeClair et al., 2019; Kim et al., 2020), or linearized as bags of AST paths (Alon et al., 2018c;a). Graph representations have been used in conjunctions with GNNs (Allamanis et al., 2017; Brockschmidt et al., 2018; Wei et al., 2020) and have been recently combined with RNNs and (relational) transformers (Hellendoorn et al., 2019b).

Yet, most of these works are evaluated on a single task, yielding limited insights on the trade-offs of various representations and models. GLUECode's goal is to ease experimentation across representation and modelling choices on a variety of local and global tasks. To achieve this, we provide several pre-processed representations at the level of source code files: raw text, tokenized code, abstract syntax trees, graph representations (as in Allamanis et al. (2017)), and bags of AST paths as in Alon et al. (2018c;a). For global context we provide project-level call graphs. Across all representations, source code entities (methods and classes) are identified via a Universally Unique Identifier (UUID), and can be linked together. Appendix A provides details and examples.

Modelling decisions have significant impact on the performance of models and many different representations are possible, especially when considering models that perform global reasoning. GLUECode tasks are defined as a mapping from the UUID of the entity of interest to its label. Researchers can build their own input representations based on how they want to solve GLUECode. This allows researchers to combine these preprocessed representations as they see fit. GLUECode provides an API to efficiently access these representations to define the model. We show examples of the representations in Appendix A.

## 2.3 DATA

Performing pre-processing at scale is very challenging and time consuming. To extract the representations and some of the labels for the tasks, we use a variety of tools. Some of these tools perform extensive static analyses, and for this reason they require code that is compilable. Automatically compiling large amounts of arbitrary code is surprisingly difficult, as some systems may have convoluted build processes, or depend on a large number of libraries that may need to be present at compile time. We restrict our scope to Java since it is a popular language, with a lot of mature projects, and extensive tool support. To ease this task, our starting point is the 50K-C dataset (Martins et al., 2018), which is a set of 50,000 Java projects extracted from GitHub, that are compilable. Of the 50,000 projects in 50K-C, many are too small to represent realistic software projects, such as projects authored by students. This is why we restrict our scope to projects that have 50 or more Java files. This leaves us with 6,925 projects, of which we were able to compile ~5,300. These projects have a combined total of 371,492 class files, and 2,361,111 method declarations. Once the projects are compiled, we run additional tools to extract all the representations, and extract some of the labels that the tasks need. Note that the entire process took several months, which we thus spare other researchers—simply trying to compile ~7,000 projects is a weeks-long endeavour. We provide additional data processing details in Appendix A.

```
int  countBlueElements(Iterator<Element> buffer) {
    int count = 0;
    for (Element  n: buffer) {
        Color col = n.getColor();
        if (col.toString() == "Blue") {
            count = count + 1;
        }
    }
    return count;
}
```

NPath:        3
Operators:    + *
Naming:       countBlueElements *
Completion:   getColor *
NullDef:      col

\* masked in code snippet while training

Figure 1: Code snippet illustrating the five tasks in GLUECode.

## 2.4    THE GLUECODE TASKS

To incentivize the community to develop models that leverage the structured and unstructured nature of code to perform global reasoning, we define several tasks that cover a spectrum in terms of reliance on the structure of code, and the need for non-local reasoning. Thus, each of the five GLUECode tasks is meant to test different reasoning capabilities of a model. An overview is shown in Table 1. We describe the tasks next and provide an extended discussion on the design of each tasks in Appendix B, including discussion of alternatives we discarded. Figure 1 shows how each task looks like in an artificial snippet. Note that global tasks may need additional context; for instance, a caller of `countBlueElements` passing a `buffer` that triggers a null dereference.

**Task Selection Rationale.** We selected five tasks: three are inspired by practical scenarios, while two have labels generated by static analyzers. Models that succeed at the Operator Prediction task may be used to spot potential bugs in existing code (Pradel & Sen, 2018); models that succeed at Method Naming may be used to provide refactoring recommendations on legacy code bases; and models that succeed at Code Completion may be integrated in an IDE's code completion engine. For the two tasks that have labels generated by static analyzers (NPath complexity and NullToken), we are not interested in merely replicating these programs. Rather, our goal is to incentivize the development of neural architectures that can demonstrate these forms of reasoning (fine-grained reasoning about the control and data flow of programs, both locally and globally), so that future models may incorporate these reasonings to succeed in more practical tasks.

**Task format and metrics.** Two tasks in GLUECode are classification tasks, while the other three other are sequence generation tasks. We initially wanted all the tasks to use the same format, for simplicity and uniformity. However, this proved too restrictive as it severely limited the tasks that we could include, or led to variants of the tasks that were too easy. The sequence generation tasks use different metrics, to more closely fit to the scenario they represent. Since all performance metrics range between 0 and 1, we simply average them to obtain an overall score for a given model.

**Unit of interest.** In all GLUECode tasks, the unit of interest is a method. Thus, for each task, the dataset is a mapping from a unique method ID to a label. As part of pre-processing, researchers can retrieve the representation they wish, including related source code entities (e.g., callers and callees of the current method). Note that we mask information that could lead to data leakage in these additional source code entities (e.g., for the method naming task, we mask the method call in the callers). To further prevent data leakage, for tasks that rely on global context, the training, validation, and test set is split at the project level, such that samples from projects in the validation and test set are unseen during evaluation. We also provide a development set.

**Size of datasets.** The size of the dataset is dictated by several factors. Overall, we are limited by the number of projects we have analyzed, as adding more projects requires a significant pre-processing effort. For tasks like Method Naming and Code Completion we have about a million samples per task. While for other tasks (e.g. NullToken), the number of available examples is limited, as the analysis is expensive to run and returns a small number of examples. For classification tasks, some classes are less common, and we take as many samples as possible across all classes to have a balanced dataset. While several other works propose larger datasets, which may be more desirable for some purposes, we note that smaller datasets have two advantages: they ease the computational burden, and incentivize the community to work towards more sample-efficient models. Moreover, other models may use the pre-training paradigm to generate convincing results with limited samples.

### 2.4.1 NPATH COMPLEXITY

NPath complexity prediction is purely structural and local: it can be solved while fully ignoring identifiers and non-local context. We used PMD to extract the NPath code complexity metric (Nejmeh, 1988), which counts the number of distinct paths control flow can take in a method. To succeed at this task, a model needs to keep track of the control structures and how they relate to each other (e.g. via nesting). It needs to do this while considering the entire scope of each method. The task is formulated as a classification task, with a balanced set of 12 complexity buckets. Note that since NPath is unevenly distributed, we use buckets that redistribute the complexity values in our dataset evenly. The target metric is classification accuracy.

### 2.4.2 OPERATOR PREDICTION

The second task involves mostly local reasoning, but in contrast to NPath complexity, it leverages both structured and unstructured information. The task consists of predicting a masked operator in the method body, similar to DeepBug (Pradel & Sen, 2018). This involves structural reasoning as the context is useful in determining the type of operators (e.g., Is the operator in an if condition?), as well on the identifier names which may embed information valuable in determining the operator type (e.g., an identifier "`maxQuantity`"). While we expect the task to mostly rely on local reasoning in the method body, non-local reasoning may be helpful too (e.g., getting type information from instance variables or method return types).

The task has 12 classes spanning the most common operators: The 5 arithmetic operators (basic operations and modulo), six Boolean comparison operators, and the assignment operator. The classes are balanced, and we use accuracy as a metric. For each method, a single operator is masked, even if there are multiple operators present in the method.

### 2.4.3 METHOD NAMING IN CONTEXT

In method naming task (Allamanis et al., 2016; Alon et al., 2018c), the method name is masked and needs to be predicted. This can be seen as a summarization task (of the method body). A model must reason over the body, both at the level of the structure (control and data flow), and at the level of identifiers, to succeed at this task.

While most existing formulations of the task have been restricted to using the method body, GLUECode does *not* impose such a restriction; indeed we expect that adding additional context, such as class-level information and information from the calling contexts, to lead to performance improvements. For instance, having access to the class context may allow a model to better leverage naming conventions of the project. Likewise, useful information may be found on method usages (invocations), such as the names or values given to the parameters or the return value. Thus, GLUECode provides the facilities to incorporate such information in models and representations. Note that to avoid data leakage, we mask the target method name in each caller's context, across representations. In contrast to traditional method naming, we use a character-level BLEU as an evaluation metric. The rationale is that is independent of tokenization (Denoual & Lepage, 2005), and reduces the weight of common, but short subwords such as "`get`" (see Appendix B for details).

### 2.4.4 CODE COMPLETION IN CONTEXT

Code completion is another task that has been used to evaluate recommendation algorithms (Robbes & Lanza, 2010) and source code models, particularly autoregressive language models (Hellendoorn & Devanbu, 2017; Karampatsis et al., 2020). We recast the task as masked language modelling task, similar to Alon et al. (2020). Having a code completion task as a masked language modelling task allows model to leverage both the preceding context and the following context, which makes the task relevant in a scenario where a programmer would be modifying existing code. Furthermore, we restrict the task to predict only method calls, not other types of tokens. This has two benefits: 1) it makes the task more challenging by removing tokens that are very easy to predict such as parentheses and semicolon, and 2) it emphasizes the tokens for which non-local reasoning is beneficial.

Since the goal is to predict a method call inside a method body, the whole project scope is relevant. While in method naming, models summarize an entire method body in a new — possibly unseen — name, in code completion, a model should identify which of the *existing* method calls fits. These

Table 1: GLUECode: Tasks at a Glance

| Task | Structure | Identifiers | Global Scope | Type | Train | Test | Dev |
|------|-----------|-------------|--------------|------|-------|------|-----|
| **NPath** | +++ | - | - | Classification | 9,600 | 1,200 | 1,200 |
| **Operators** | ++ | ++ | +/- | Classification | 9,600 | 1,200 | 1,200 |
| **Naming** | ++ | ++ | + | Generation | 800K | 100K | 100K |
| **Completion** | + | +++ | ++ | Generation | 800K | 100K | 100K |
| **NullToken** | +++ | + | +++ | Generation | 10K | 1K | 1K |

methods could be defined in the same class, in another class or package in the system, or imported from a dependency. This makes the method completion task much more amenable to performance improvements when the non-local context is taken into account.

For this task, GLUECode uses exact match accuracy: models should generate the exact masked token. Unlike method naming, a close match does is not valid (in a practical scenario, a close match would likely result in an error). The call graph representation of the system hides any links between the target and the called method, to avoid data leakage.

### 2.4.5 NULL DEREFERENCE PREDICTION

The last task is null dereference prediction. This task should benefit the most from non-local reasoning. To succeed at this task, models should be able to reason across the control flow and the data flow of several methods at once. For this task, we use the Infer static analyzer (Facebook, 2015) to find possible null dereferences. Infer performs full-program static analysis to track the possible values of variables, and emits warnings when it finds a possible execution path in which a null pointer dereference can occur. These execution paths can span several methods, across several files, and point to the line number and exact token in which the null dereference can occur. This kind of reasoning *requires* non-local reasoning for most of the warnings emitted by Infer (except those where the execution path that was found does not exit the method body). We ran Infer on all the projects in the dataset. Since Infer's analysis is precise, it does not produce many warnings (~20,000 in total), unlike other static analysis tools such as FindBugs (Ayewah et al., 2008) which are more prone to false positives.

The goal of the task is to output the token where the null dereference may occur. Similar to code completion, we measure accuracy, considering only exact matches. We also added 20% of negative examples, in which the model has to output a special token signifying that no null dereference warning could be found, to incentivize models to account for this eventuality. Thus, a naive baseline always predicting this token would have a maximum accuracy of 20%.

## 3 EVALUATION

We provide performance results for several simple baselines (MLPs, LSTMs and CNNs), as well as a more advanced model: a pre-trained transformer. All these models perform local reasoning, and treat the source code as a sequence of tokens. There are, of course, many more advanced models that could be evaluated on GLUECode, starting with models that are limited to local reasoning but also exploit source code's structure, such as Tree-LSTMs, linearized ASTs, or Graph Neural Networks. The space of possibilities grows even further if we consider models that incorporate non-local reasoning; if not, there would not be a need for GLUECode in the first place. Thus, the baselines we provide should be taken as a starting point, giving insights on the lower bound exhibited by simple baselines, as well as the performance of a pre-trained transformers that are closer to the state of the art. Significant exploration of the performance of models lies ahead, a task for which we welcome the involvement of the community.

**MLP.** A simple Multilayer Perceptron with a single hidden layer, intended to represent a very simple but non-naive baseline. The input embedding layer has a maximum size of 200 tokens. The single dense hidden layer has 64 hidden units. The output layer is a softmax layer over the all the classes for classification, or the entire vocabulary for the generation task.

Table 2: GLUECode results across baselines

| Model | NPath | Operators | Naming | Completion | NullToken |
|-------|-------|-----------|--------|------------|-----------|
| **MLP** | 0.326 | 0.357 | 0.169 | 0.288 | 0.344 |
| **BiLSTM** | 0.372 | 0.465 | 0.221 | 0.480 | 0.316 |
| **CNN** | 0.331 | 0.297 | 0.194 | 0.451 | 0.296 |
| **Seq2Seq/Seq2Tok** | 0.543 | 0.368 | 0.262 | 0.524 | 0.228 |
| **Transformer** | 0.747 | 0.782 | 0.389 | 0.534 | 0.239 |

**CNN.** A Convolutional Neural Network, with an embedding layer, followed by a 1D convolution layer of size 5, and by a global average pooling layer. These are followed by a dense hidden layer and an output layer similar to the MLP above. We use it to explore the impact of the inductive bias of convolution on the GLUECode tasks.

**BiLSTMs** A Bidirectional sequential model, where the embedding layer is followed by a single bidirectional LSTM layer, a dense layer and the output layer. It also uses a softmax layer for all tasks (predicting tokens over all the vocabulary for sequence generation tasks).

**Seq2Seq/Seq2Tok** Another LSTM variant that uses a unidirectional encoder-decoder architecture and predict tokens as sequences of camelCase-separated subtokens (Seq2Seq), or a single token for the classification tasks (Seq2Tok). Both variants allow us to explore the impact of the sequential inductive bias. Seq2Seq and Seq2Tok allow us to reduce the impact of OoV tokens as we use subtokens.

**Transformer.** We include a stronger baseline, a Transformer, to explore the impact of the popular NLP pre-training then fine-tune paradigm. CodeBERTa is a pre-trained, 6-layer Transformer trained on the CodeSearchNet challenge dataset (Husain et al., 2020) by HuggingFace. We fine-tune it separately on each task. We chose this as our stronger baseline since pretrained transformers for code have performed very well on other tasks (Kanade et al., 2020)

### 3.1 RESULTS

The baseline evaluation results on the GLUECode tasks are presented in Table 2 above.

Overall, we see that the Transformer exhibits higher performance on the first four tasks (NPath prediction, Operator prediction, Method naming), but is only having reasonably acceptable performance on the first two tasks (Npath prediction and Operator prediction), which are the most local ones. For the tasks which have some globalness aspect to it, the transformers have an average accuracy of 40% with highest score being barely above the fifty percent threshold for the method call completion task. Even in the local tasks, where the transformers score well, there is still a margin for improvement of more than 20%.

It is important to note here that unlike method naming, completion task has many labels (method api calls) which belong to the Java standard library, such as println(), toString() etc. which are commonly used, and which are easier to predict for DL models (Hellendoorn et al.,2019a). About 20% of the dataset consist of standard library method calls. This might explain why the models perform better in comparison solely against the method naming task.

We suspect that we may have over-sampled API methods, which are easier to predict for DL models. We are considering making the task more challenging by using stratified sampling, to force the sample to have more locally defined methods than it has now.

## 4 DISCUSSION

**There is ample room for improvement.** Our goal was to provide tasks that are challenging for models that employ only local reasoning. None of the models have high performance across all the tasks; struggling on most tasks. While we expect state of the art structured models (e.g., using ASTs or graphs) to perform better on the tasks requiring mostly local reasoning, we do not except that they will reach acceptable performance on the tasks that require non-local reasoning.

**Incorporating non-local reasoning.** Significant improvements are required to develop models that better handle more global context. We expect that simple solutions such as growing models to accommodate more context will hit diminishing returns as the size of the input grows considerably. Better strategies will need to be devised.

**Impact of inductive bias.** On some tasks, the performance of the models vary widely. We hypothesize that the inductive bias of some of the models is not a good fit for some task. For instance, the Transformer trained with the MLM objective works very well for operator prediction (even without fine-tuning!), but the MLP outperforms it on the NullToken task.

**Multi-task models.** While a longer-term goal is to define multi-task models that perform well on all the tasks in the benchmark, the tasks proved challenging enough that we expect most short-term development should be geared towards single-task performance first.

## 4.1 LIMITATIONS OF THE BENCHMARK

**Additional software characteristics.** With GLUECode, we focus on two principal characteristics of software: the fact that it is structured, and that non-local reasoning is necessary. There are other characteristics we didn't take into account, such as the prevalence of natural language comments (Allamanis et al., 2015b), the fact that code can be executed (Wang, 2019), or that it evolves (Hoang et al., 2019). Other benchmarks or an extension of GLUECode would be needed to account for this.

**Comparison with previous work.** Some of our tasks (code completion and method naming) exist in previous work. While comparing with the literature would be insightful, it is difficult, as our task formulation (and our dataset) are quite different.

**Shortcuts.** Deep learning models can take shortcuts and exploit spurious correlations if they are present in the data (Geirhos et al., 2020). We spent considerable time iterating on the task selection and formulation to avoid these issues (particularly on the Nulldef task), by thoroughly investigating when our baselines had suspiciously high performance. However we cannot guarantee we have found all issues.

**Choice of metrics.** We tried to select metrics that present a fair view of performance, at the expense sometimes of reformulating a task (e.g. for method naming). When using accuracy, we were careful to balance the datasets.

**Limited number of baselines.** Our principal focus in this work is the definition of the tasks. We have a limited number of baselines that we include as a result. We plan to evalaute more baselines in future work, and we invite the community to contribute.

**Code duplication.** Code duplication is known to be extensive in software (Allamanis, 2019). A simple approach that filters out duplicated code would not work in our case, as it would make the projects to be incomplete for global contexts. We ensured that the methods in the test set are not seen in the training set, but it is possible that a handful of methods are duplicated, with unknown effects.

## 5 CONCLUSION AND FUTURE WORK

We introduce GLUECode, a benchmark for source code machine learning models that emphasizes that code is composed of interacting entities and has a fundamental structured nature. The GLUECode tasks include both tasks that require *local* and *global* reasoning, to account for source code's interacting entities. Moreover, to facilitate experimentation on range of structures, includes an exhaustive set of pre-processed source code representations (textual, ASTs, graphs) that researchers are free to leverage when they are building their models. The data collection and preprocessing for the task datasets and generating multiple representations for each data sample, scaled at the size of thousands of projects, took several months, which we spare the community. We also tested several baselines, ranging from simple neural models to pretrained transformers. The results indicate that there is a lot of progress to be made on the GLUECode tasks. The design space of models that leverage global reasoning on complex, structured data is even larger than for local models. Thus, we invite the community to download our code representations, write "glue code" to transform these representations as they see fit, and evaluate the resulting source code models on GLUECode tasks.

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

## A APPENDIX: ADDITIONAL DETAILS ON THE DATASET AND REPRESENTATIONS

### A.1 THE 50K-C DATASET

The projects in 50K-C (Martins et al., 2018) where harvested from GitHub, and selected as they included a build script which made automated compilation of the dataset available. We need compilable projects as additional post-processing tools require Java bytecode to work. However, many of the projects are small, so we selected the ∼7,000 projects with 50 or more classes, as a proxy for more mature projects. While trying to compile the projects, we did notice some failures, mainly related to some unresolved libraries. Since we had still ∼5,300 projects that compiled successfully, we did not investigate it further. We use Andrew Rice's feature graph extractor (https://github.com/acr31/features-javac) to extract feature graphs similar to the ones in Allamanis et al. (2017), but for Java instead of C#. This representation allows us to also extract the AST and token representations, by simply omitting unnecessary edges. Note that compiling projects and extracting feature graphs both took several weeks to simply execute.

Of note, these feature graphs are at the file level, not the project level. We thus use the Java call graph extractor (https://github.com/gousiosg/java-callgraph) of Georgios Gousios to extract inter-procedural call graphs. We then link the entities across representations using their UUIDs, and apply further post-processing to disambiguate some method calls between file. In the cases where a method call can not be disambiguated (e.g., a polymorphic method call), we include all possible edges in the call graph.

### A.2 AVAILABLE REPRESENTATIONS IN GLUECODE

Here, we present the code representations readily-available with our benchmark. We choose a data sample from our dataset, and present the same data sample in various representations. Based on machine learning model, different representations corresponding to the same data samples are readily available making evaluation on tasks versatile across different model types. All representations are stored in a database, where they are accessible via a sample's UUID.

**Raw Text** The first text representation we have for every data sample is the raw text. Each line is comma separated, and even the line breaks and tab spaces are preserved.

```
public static Key getKey(String ahex)
,  {
,    try
,    {
,      byte[] bytes = CHexString.toByteArr(ahex);
,      SecretKeySpec skeySpec = new SecretKeySpec(bytes, "AES");
,      return skeySpec;
,    }
,    catch( Exception e )
,    {
,      System.err.println("CAesEncrypt.getKey: " + e);
,      return null;
,    }
,  }
```

**Tokens** The second representation is the list of method tokens which are ready to use, or further pre-processed if a model using subword units is desired.

```
PUBLIC,STATIC,Key,getKey,LPAREN,String,ahex,RPAREN,LBRACE,TRY,LBRACE,byte,
LBRACKET,RBRACKET,bytes,EQ,CHexString,DOT,toByteArr,LPAREN,ahex,RPAREN,SEMI,
SecretKeySpec,skeySpec,EQ,NEW,SecretKeySpec,LPAREN,bytes,COMMA,"AES",RPAREN,
SEMI,RETURN,skeySpec,SEMI,RBRACE,CATCH,LPAREN,Exception,e,RPAREN,LBRACE,
System,DOT,err,DOT,println,LPAREN,"CAesEncrypt.getKey:",PLUS,e,RPAREN,SEMI,
RETURN,null,SEMI,RBRACE,RBRACE
```

**AST** We also have AST representation of every data sample, where the *ast_labels* are the list of nodes of the data sample, and *ast_edges* are the list of tuples with parent-child edges.

```
{
    "ast_labels": ["METHOD", "NAME", "MODIFIERS", "FLAGS", "RETURN_TYPE",
        "IDENTIFIER", "NAME", "PARAMETERS", "VARIABLE", "NAME", "TYPE",
        "IDENTIFIER", "NAME", "BODY", "BLOCK", "STATEMENTS", "TRY",
        "BLOCK", "STATEMENTS", "VARIABLE", "NAME", "TYPE", "ARRAY_TYPE",
        "TYPE", "PRIMITIVE_TYPE", "PRIMITIVE_TYPE_KIND", "INITIALIZER",
        "METHOD_INVOCATION", "METHOD_SELECT", "MEMBER_SELECT",
        "EXPRESSION", "IDENTIFIER", "NAME", "IDENTIFIER", "ARGUMENTS",
        "IDENTIFIER", "NAME", "VARIABLE", "NAME", "TYPE", "IDENTIFIER",
        "NAME", "INITIALIZER", "NEW_CLASS", "ARGUMENTS", "IDENTIFIER",
        "NAME", "STRING_LITERAL", "IDENTIFIER", "NAME", "RETURN",
        "EXPRESSION", "IDENTIFIER", "NAME", "CATCHES", "CATCH", "BLOCK",
        "STATEMENTS", "EXPRESSION_STATEMENT", "EXPRESSION",
        "METHOD_INVOCATION", "METHOD_SELECT", "MEMBER_SELECT",
        "EXPRESSION", "MEMBER_SELECT", "EXPRESSION", "IDENTIFIER",
        "NAME", "IDENTIFIER", "IDENTIFIER", "ARGUMENTS", "PLUS",
        "LEFT_OPERAND", "STRING_LITERAL", "RIGHT_OPERAND", "IDENTIFIER",
        "NAME", "RETURN", "EXPRESSION", "NULL_LITERAL", "VALUE",
        "PARAMETER", "VARIABLE", "NAME", "TYPE", "IDENTIFIER", "NAME"],
    "ast_edges": [
        [0, 1],
        [0, 4],
        [0, 7],
        [0, 13],
        [0, 2],
        [2, 3],
        ...
        [54, 55],
        [55, 81],
        [55, 56],
        [56, 57],
        ...
        [79, 80],
        [81, 82],
        [82, 83],
        [82, 84],
        [84, 85],
        [85, 86]
    ]
}
```

**Code2Vec** We have Code2Vec representations for every data sample. Each method is represented as a set of up to 200 AST paths; in case the method has more than 200 possible paths, the 200 paths are selected at random. Each path is a combination of AST node labels, represented as a unique symbol.

```
get|key key,362150388,getKey key,714300710,ahex
    key,-1248995371,string getKey,-1103308019,ahex
        getKey,1228363196,string
    ...
    e,-850278433,println e,910578178,null println,-1488546123,null
```

**Code2Seq** We also have Code2Seq representations for the entire dataset of samples. These are similar to Code2Vec representations, but the identifiers are sequences of camelCase-separated tokens, while the paths are sequences of AST node labels.

```
get|key key,Cls0|Mth|Nm1,getKey key,Cls0|Mth|Prm|VDID0,ahex
   key,Cls0|Mth|Prm|Cls1,string getKey,Nm1|Mth|Prm|VDID0,ahex
   getKey,Nm1|Mth|Prm|Cls1,string
   ...
   e,Nm1|Plus2|Cal|Nm3,println e,Nm1|Plus2|Cal|Ex|Bk|Ret|Null0,null
      println,Nm3|Cal|Ex|Bk|Ret|Null0,null
```

**Feature Graphs** Finally, we have the feature graph representation for each sample of the dataset. The *node_labels* key lists all nodes in the feature graph, while the *edges* key has information about every edge type and the corresponding connections.

```
{
   "backbone_sequence": [13, 14, 15, 16, 17, 18, 19, 20, 21, 22],
   "node_labels": ["METHOD", "NAME", "MODIFIERS", "FLAGS",
      "RETURN_TYPE", "IDENTIFIER", "NAME", "BODY", "BLOCK",
      "STATEMENTS", "RETURN", "EXPRESSION", "STRING_LITERAL", "PUBLIC",
      "String", "METH_PLACEHOLDER", "LPAREN", "RPAREN", "LBRACE",
      "RETURN", "\"Login request processing\"", "SEMI", "RBRACE"],
   "edges": {
      "CH": [
         [0, 1],
         [0, 4],
         [0, 7],
         [0, 2],
         [2, 3],
         [4, 5],
         [5, 6],
         [7, 8],
         [8, 9],
         [9, 10],
         [10, 11],
         [11, 12]
      ],
      "NT": [
         [13, 14],
         [14, 15],
         [15, 16],
         [16, 17],
         [17, 18],
         [18, 19],
         [19, 20],
         [20, 21],
         [21, 22]
      ],
      "LU": [],
      "LW": [],
      "CF": [],
      "LL": [],
      "RT": [],
      "FA": [],
      "GB": [],
      "GN": []
   },
   "method_name": ["get", "Servlet", "Info"]
}
```

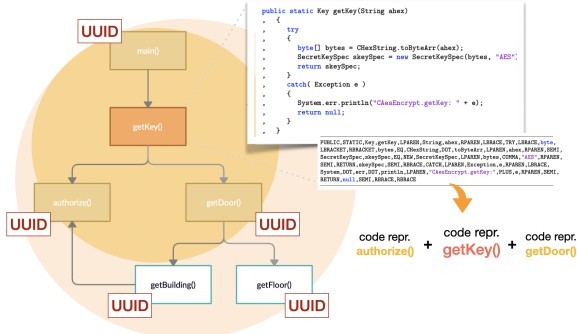

Figure 2: Illustration for global context

## A.3 COMBINING REPRESENTATIONS FOR GLOBAL CONTEXT

For global context we provide project-level call graphs. Across all representations, source code entities (methods and classes) are identified via a Universally Unique Identifier (UUID), and can be linked together.

For every project, we provide a call graph representation of the entire project. This representation is a graph where the nodes are method UUIDs, and the edges represent caller/callees relationships. This representation can be used to retrieve callers and callees of the method of interest, or even the entire project's call graph, should researchers wish to do so.

# B    APPENDIX: ADDITIONAL DETAILS ON THE GLUECODE TASKS

## B.1    NPATH

We used the PMD static analyzer to compute the NPATH complexity of the methods in the dataset. PMD implements a variety of static analysis checks. The detailed description of the NPATH complexity metric, as implemented in PMD, is available at `https://pmd.github.io/latest/pmd_java_metrics_index.html#npath-complexity-npath`. Of note, NPATH grows exponentially, as consecutive statements have their complexity multiplied. This can lead to very high NPATH values. The distribution of the metric is highly skewed, with many more methods that have low complexity values than ones with higher ones. In addition, there are peaks in the distribution as values that are powers of two are more numerous than others. As a result, we defined variable size bins to have an appropriately balanced dataset. Our bins are 1,2,3,4,5-6,7-8,9-10,11-15,16-20,21-30,31-50,51-100

**Alternatives we considered.**   We considered several other tasks that incentivize structure at the local level, such as tasks that would involve replicating local static analyzes. We considered having four tasks representing each canonical local static analyses: Live variables ("backwards may"); Reaching definitions ("forwards may"); available expressions ("forwards must"); and very busy expressions ("backwards must"). However, we felt this would have weighted too heavily on local tasks, hence we decided for a single task. We had considered other common complexity metrics such as Halstead's complexity metrics and McCabe's cyclomatic complexity, and we prototyped a version of this task using McCabe's complexity. Ultimately, we decided against it, as it did not require models to reason on how control flow statements relate to each other; it was limited to counting operators.

## B.2    OPERATOR PREDICTION

Since not all operators are equally rare, we made choices among the most common operators, in order to have a balanced dataset in the end. We also had to select operators that could be plausibly mistaken from one another, leading us to discard additional operators. We ended up choosing the following operators: ``+``, ``-``, ``*``, ``/``, ``%``, ``=``, ``==``, ``!=``, ``<``, ``>``, ``<=``, and ``>=``. Thus, we have two larger classes of arithmetic operators on the one hand, and boolean operators on the other. We find that models do pick up on this, and tend to missclassify arithmetic operators with other arithmetic operators, and boolean operators with other boolean operators.

**Alternatives we considered.**   We considered other tasks that, similarly to operator prediction, were mostly local but were more "holistic" in their reasoning. An early candidate was the "VarMisuse" task of (Allamanis et al., 2017), where models have to detect whether a variable is replaced by another, type-compatible variable. However, this requires extensive static analysis, that is so far only implemented for C#, not Java. We also considered other "Misuse" variants, such as an "OperatorMisuse" variant of operator prediction. We decided against this as we were concerned that substituting an operator with another may turn out to be too easy of a task, and that models may take shortcuts in their reasoning. An interesting other task would be predicting the output of programs, as in (Zaremba & Sutskever, 2014); this would however diverge from our goal, as the task involves generated code snippets.

## B.3    METHOD NAMING

We initially considered all the methods in the corpus, after accounting for code duplication. We did find that a significant number of methods had very short names, which inflated performance on the task. Thus, we filtered out most method names that were shorter than 4 characters; we left a small portion of them (around 23,000) in order to arrive at a round number of one million method names. We use the character-level BLEU metric described in Denoual & Lepage (2005), with smoothing "Smoothing1" from (Chen & Cherry, 2014). We replace the method name with a special mask token, also replacing it in the method body (in case the method is recursive or forwards it to a similar, or

uses `super`, and also replacing it in the callers of the method, for models that want to use those in their global reasoning.

**Alternatives we considered.** We considered other tasks that involve reasoning over the whole method body, such as a summarization variant in which the task is to predict a method comment (such as in (LeClair et al., 2019). This task had the advantage of also requiring models to generate natural language, but we felt this complexified the architecture on the decoding side, and would dillute the focus of the benchmark. We also considered clone detection tasks (Mou et al., 2016; Wei & Li, 2017), but these would require the models to reason over a pair of entities, which would also complexify the models for a single task (a more drastic change, as it is on the encoder side).

We also had extensive discussions on the metric to use. The state of the art evaluates method naming by tokenizing the prediction and the target according to camelCase convention. This has two disadvantages: 1) it adds a bias towards models that tokenize identifiers in the same way (while recent models tend to use variants of byte-pair encoding (Sennrich et al., 2015), that may not respect the camelCase convention), and 2) it weights common subwords such as "`get`", "`set`", or "`is`" too heavily, distorting performance. We instead use a character-level BLEU metric that is independent of the tokenization (Denoual & Lepage, 2005), and reduces the weight of these common, but very short subwords. This allows researchers to experiment with the tokenization that they prefer, and makes the task more challenging while still rewarding close, but not exact matches (e.g., similar words but with different endings). We also considered other character-level metrics, such as the Jaro-Winkler string distance (Winkler, 1990). However, we found that it had a "high floor", giving relatively high scores to very distant guesses, and emphasizing similarities in the prefix, which increased the weight of the easy subwords; both issues made it harder to accurately measure progress on the task.

### B.4 METHOD COMPLETION

In each method in the dataset (the same one as method naming), we mask a single method call in the method body, at random. The task is to predict this token, with only exact matches allowed: a code completion engine that would recommend "near misses" would not be very useful. The method call could be to a method in the same class, to a method in a different class in the same java package, to a method anywhere in the system, or to a method imported from a library. Each of these cases involves different kinds sizes of context and different kinds of reasoning. Models leveraging only local reasoning will have to generate identifiers from scratch, increasing the probability of these "near misses". Models that use global reasoning could, on the other hand, learn to copy an identifier in the extended context. Existing work show that deep learning with local reasoning can be more successful in predicting API method calls (more likely to be seen in training) than method calls found in the project (Hellendoorn et al., 2019a). Beyond masking the method call token, we also mask call edges to the method that might be present in other representations.

**Alternatives we considered.** While looking for tasks that involve local masking of the method body, but would require models to take into account global context, a very close second alternative we considered was type prediction, for which a few more global models already exist (Wei et al., 2020; Allamanis et al., 2020). We ultimately preferred method call completion as the set of potential candidates (methods) is larger and finer grained than in type prediction (classes). We also discussed variants of method call completion, namely whether to ask models to hide and complete the arguments to the method call, as is done in (Alon et al., 2020). However, completing the arguments to the method call would have increased the weight of the local context, as most arguments are variables defined in the context. This would have made the task less aligned with the benchmark's goal.

### B.5 NULLTOKEN

For each warning, Infer produces a report that contains: an error message, the line number where the null dereference happens, and a trace of abstract interpretation steps that Infer took to find the potential null dereference. This trace ranges from simple, local cases (e.g., taking a particular if branch while a variable is not yet initialized), to highly complex cases covering dozens of steps across multiple methods, scattered over several files. Over all the projects, infer took several weeks to execute, and produces on the order of 20,000 warnings, showing that these warnings are pretty

rare. We did filter some of the warnings: some methods had more than one warning, which would make the task ambiguous for the models, so we discarded all warnings in this case.

**Alternatives we considered.**   Infer (Facebook, 2015) has several precise, interprocedural analyses that are strong candidates for tasks that require precise modelling and reasoning over multiple entities. Examples include reachability analysis (finding whether method A can call method B, directly or indirectly), or an analysis that estimates the runtime cost of a method (including the cost of methods that it calls). All of these tasks have the drawback that we are asking the model to emulate the reasoning of an existing tool. One of the deciding factors was that Null dereference detection, while being a task that requires us to emulate the reasoning of a tool, is closer to a practical scenario, as it provides warnings for real bugs. Another alternative in that area would be to use a Taint analysis tool, such as (Arzt et al., 2014); however, we would expect that taint analysis warnings would be even rarer than possible null dereferences.

We initially tried a simpler version of the task, which was a straightforward binary classification at the method level (whether there a null dereference warning in this method), with a balanced sample of positive and negative methods. However, selecting negative examples proved to be difficult, as even simple models found spurious correlations that led to inflated performance in this simplified version of the task. We thus settled for a generation version of the task, where the goal is to output the token in which the null dereference can occur. We also discussed the amount of negative examples to include, finding that 20% was a reasonable tradeoff, that required models to envision that having no null dereference was a possiblity, while not inflating disproportionately the performance of trivial baselines that always predict this label.

We also considered more complex version of the task, such as requiring models to predict steps in Infer's execution traces, but we thought they might prove too difficult at this time. We also considered a variant where the model would need to predict the line number (starting from the beginning of the method) instead of the actual token, but didn't choose this since task would then become sensitive to code formatting choices.

## C  APPENDIX: DETAILS ON THE BASELINES

**Vocabulary** MLP, CNN and BiLSTM all use a full-token vocabulary of 10,000 elements, initialized on the training set of each task. Tokens that are not in the top 10,000 are replaced by OOV tokens. Seq2Seq splits token via the camelCase coding convention to reduce vocabulary size, while the pretrained Transformer uses it's original open vocabulary (using Byte-Pair encoding).

**MLP:** A model with an embedding layer of vocabulary size 10,000, embedding dimension 64, and input maximum length 200, as its first layer. This converts our words or tokens into meaningful embedding vectors. This is fed into a single, dense hidden layer of size 64. We use ReLU as our activation function. The output layer has a softmax activation. We compile the model with the Adam (Kingma & Ba, 2014) optimizer, and use *sparse categorical cross-entropy* as our loss since we are going to use the same model for classification and generation (this models treat generation as classification over the entire vocabulary).

**BiLSTM:** A model with an embedding layer of vocabulary size 10,000, embedding dimension 64, and input maximum length 200, as its first layer. This converts our words or tokens into meaningful embedding vectors. Then we add our Bidirectional LSTM layer. The standalone LSTM layer is initialized with the value of the embedding dimension. The LSTM layer is then wrapped with a Bidirectional layer wrapper. We then add a densely-connected neural network layer on top of that with the number of units equal to the embedding dimension, and use ReLU as our activation function. And yet another layer, with softmax activation, which is our output layer. We compile the model with the Adam (Kingma & Ba, 2014) optimizer, and use *sparse categorical cross-entropy* as our loss since we are going to use the same model for multi-class classification.

**Seq2Seq/Seq2Tok:** Same as BiLSTM, but is unidirectional with an encoder/decoder architecture and uses camelCase-separated tokens, reducing OOV.

**CNN:** For our base CNN model, use an embedding layer of vocabulary size 10,000, embedding dimension 64, and input maximum length 200, as our first layer. We then add a 1D convolution layer, specifying the dimensionality of the output space 128, the size of 1D convolution window 5, and the activation function which we set to ReLU. We then add a 1D global average pooling layer to reduce the data dimensionality, so as to make our model faster. The last two layers on top of the pooling layer are identical to our LSTM model, we add a densely-connected neural network layer with the number of units equal to the embedding dimension, and use ReLU as our activation function. We then add another dense layer as our output layer, with a softmax activation.

We also choose *sparse categorical cross-entropy* as our loss function as we use the same model for all the tasks. We compile the CNN model with the Adam Kingma & Ba (2014) optimizer.

**Transformer:** We use `CodeBERTa-small`[1], a pre-trained, 6-layer transformer based on the *RoBERTa* (Liu et al., 2019) architecture. The model was pre-trained on 2 million functions written in six different languages (including Java) from the *CodeSearchNet* dataset(Husain et al., 2020) and released by Huggingface (Wolf et al., 2020).

---

[1]`https://huggingface.co/huggingface/CodeBERTa-small-v1`

# D    APPENDIX: RELATED WORK

## D.1    BENCHMARKS

Many communities create benchmarks to advance the state-of-the-art of their field. Arguably, the ImageNet challenge (Russakovsky et al., 2014) is one of the most well-known benchmarks in the machine learning and computer vision community. In software engineering, Sim et al. (2003) urged to adopt benchmarking as an evaluation measure, based on the impact it has on community building. While in the performance community, benchmarks such as the one from Blackburn et al. (2006) have been used. Below we provide a brief overview of some NLP benchmarks, as an extended related work, which focus beyond a single task.

**bAbI Tasks** Weston et al. (2015) present several NLP tasks in simple question-answering format intended to test dialogue agents on natural language understanding. bAbI aimed to provide a yardstick for researchers to assess their NLP models for intelligent dialogue agents. The tasks in bAbI are artificial, but measure specific aspects of reading comprehension, such as reasoning by chaining facts, simple induction, deduction, etc., and have well-defined degrees of difficulty.

**GLUE Benchmark** To progress towards the generalizability of NLP models, Wang et al. (2018) present the GLUE benchmark to evaluate and analyze the performance of NLP models across a diverse range of existing tasks. They further evaluate baselines for multi-task and transfer learning, comparing them to training a separate model per task.

**SuperGLUE Benchmark** With the performance of NLP models on the GLUE benchmark surpassing the level of non-expert humans, Wang et al. (2019) reinforce their GLUE benchmark by presenting the SuperGLUE benchmark with harder tasks and more diverse task formats.

**DecaNLP Benchmark** Going beyond the paradigm of task-specific NLP models, McCann et al. (2018) present a set of ten tasks, to evaluate general NLP models. They cast all tasks in a Question-Answering format over a given context, and present their own Multitask Question Answering Network (MQAN) that jointly learns on all tasks.

## D.2    CODE PROBLEM TASKS

Here we detail some related problem tasks in the source code domain, for machine learning source code models. Several studies have worked on source code-related tasks (Allamanis et al., 2018), some of which we discuss here. These tasks are examples of problem tasks we could address to a great degree with the aid of modern deep learning methods.

**MethodNaming** A machine learning model of source code aims to predict the name of a certain method, given its code body. This problem task was explored by multiple studies (Allamanis et al., 2015a; 2016; Alon et al., 2018a; Fernandes et al., 2018).

**VarMisuse** This goal of this task is to detect and fix incorrect variable uses within a program. Given the source code, a machine learning model should determine if a certain variable has been misused at a given location. For example, a developer, might use `i` instead of `j` in an index. Allamanis et al. (2017); Hellendoorn et al. (2019b) addressed this task and showed that a graph neural network learns to reason about the correct variable that should be used at a given program location; they could also identify a number of bugs in mature open-source projects.

**Defect Prediction** Finding a broader set of defects in source code is another task with the potential to be extremely useful. Pradel & Sen (2017) address the problem of defect prediction by training a deep-learning based model that can distinguish correct from incorrect code. They present a general framework for extracting positive training examples from a code corpus, make simple code transformations to convert them into negative training samples, and then train a model to indicate one or the other.

**Clone Detection** This tasks deals with the identification of code clones. With available pairs of code fragments, a source code model should be able to indicate whether the sample pairs are clones. White et al. (2016) utilize a deep learning approach for the classic task of code clone detection, both at the file and the method level with promising results.

### D.3 SOURCE CODE REPRESENTATIONS

Representing source code for the consumption in machine learning models is an active research area. In the recent past, programs were generally represented as a bag of tokens to be fed into machine learning models, but multiple studies (Allamanis et al., 2017; Alon et al., 2018a;b; Maddison & Tarlow, 2014) have now shown that leveraging the structured nature of source code helps machine learning models to reason better over code; and the models trained on such representations perform consistently well over sequential or less-structured program representations. Therefore, in our discussion here we include program representations which make use of some form of program structure, whether by extracting information from abstract syntax tress, control-flow or data-flow graphs, or similar structures.

**AST** The abstract syntax tree (AST) is one of the most commonly used structured representation for code. There are multiple ways to exploit this structure. Some studies directly model the AST as a sequence of applications of a context-free grammar (Bielik et al., 2016; Maddison & Tarlow, 2014), and augment the grammar with long-range information (Yin & Neubig, 2017; Brockschmidt et al., 2018). Various other approaches have considered "summarizing" the tree-like structures recursively, inspired from work in NLP. For example, Büch & Andrzejak (2019) use the AST node type and node content to create node representations of a function. Mou et al. (2016) use a convolutional architecture on ASTs.

More recently, Alon et al. (2018b;a) linearize an AST into a bag of AST paths. By sampling paths from one leaf node to another, they generate a set of these paths. Finally, they use representations of the paths for the task of MethodNaming as code summarization, and code captioning.

**Path-based Embedding of CFGs** DeFreez et al. (2018) utilize inter-procedural control flow graphs (CFG) to generate function embeddings for code. They consider paths from random walks on the inter-procedural control flow graph of a program to generate the embeddings. They then use the embeddings, for C code, to detect function clones.

**Feature Graphs** Allamanis et al. (2017); Fernandes et al. (2018); Raychev et al. (2015) combine information from multiple sources, such as token sequences, ASTs, control-flow, data-flow graphs etc. of a program to generate feature graphs, which consider long-range dependencies and the structural nature of source code, to reason over source code. To learn from these graphs, these works use methods such as conditional random fields (CRF) and graph neural networks (GNN).

