# OpenReview forum: "GLUECode: A Benchmark for Source Code Machine Learning Models"
_ICLR.cc/2021/Conference — Reject_

### Official Review · AnonReviewer4 · 2020-10-26
**GLUECode: A Benchmark for Source Code Machine Learning Models**

**Rating:** 4
**Confidence:** 4

**Review:**

This paper presents GLUECode, a benchmark for evaluating machine learning models of source code. GLUECode considers both global and local contexts of source code, and aims to help researchers experiment with multiple source code representations and evaluate their models. The authors also presented results of several baselines on the benchmark.

Machine learning for source code has attracted a lot of interests in recent years. It is good to see a benchmark consists of 5000+ projects, which could help advance this area of research. The authors also performed some GLUECode tasks and presented results for several baselines, which show that there is ample room for progress on GLUECode. Overall, the paper is well written.

Concerns:

The proposed work considers both global and local contexts of code (the benchmark’s name is Global and Local Understanding Evaluation of Code). Section 2.1 also dedicates to this. However, it is not clear what global context is considered and how it is incorporated by the benchmark. In a ML for SE work, researchers may use various global contexts such as UML diagrams, library/API dependency, inter-procedural data/control flow, commit data, etc. It is not clear how these global context information can be satisfied by the benchmark.

The authors can also describe more about the unique advantages of using the proposed benchmark. Currently, they are already many public datasets released by various papers in this field (thanks to the open science policy). Also, it is easy for researchers to download a large amount of source code from open source websites (such as Github) themselves. They can also process the source code using existing static analysis tools to obtain the data they need and share the data.

Currently, GLUECode only provides a few types of source code representations. In recent years, researchers have proposed many different ways of representing source code tokens and ASTs. As an example, the following works use different AST-based source code representations (and it is not clear if the benchmark could provide necessary information to support these representations):
Yao Wan, Zhou Zhao, Min Yang, Guandong Xu, Haochao Ying, Jian Wu, and Philip S. Yu. Improving automatic source code summarization via deep reinforcement learning. In ASE, pages 397–407. ACM, 2018.

J. Zhang, et al., A Novel Neural Source Code Representation based on Abstract Syntax Tree, In Proc. the 41th International Conference on Software Engineering (ICSE 2019), Montreal, Canada, 2019.

The data quality should be discussed in detail, as low quality data will bias the analysis results. This is particularly important for a public benchmark. For example, if the benchmark contains a lot of duplicated code, the follow-up analysis will be misleading. Furthermore, software evolves. Very soon, new versions/commits will emerge. It is not clear if the evolution will degrade the data quality and the validity of the benchmark.

The proposed benchmark data and code are not available for replication purpose.

In Table 2, the baseline result for Transformer-based method completion is missing.

The paper is generally well-written. There are a few typos. For example:

In page 3, ”?? provides details and examples...”

---

> ### Author Response · Authors · 2020-11-13
> **Response to concerns Pt.II**
>
> ****
>
> **Concern 4:** The data quality should be discussed in detail, as low quality data will bias the analysis results. This is particularly important for a public benchmark. For example, if the benchmark contains a lot of duplicated code, the follow-up analysis will be misleading. Furthermore, software evolves. Very soon, new versions/commits will emerge. It is not clear if the evolution will degrade the data quality and the validity of the benchmark.
>
>
>
> > Clarification: We have carefully checked all of our datasets and can ensure that there is no duplicated code between the training and test sets. For two of the tasks with large number of samples, we even went a step further to ensure that the datasets are project-balanced, meaning that the test set only contains samples from projects not used in the training set and there is no duplicated code even for a large number of samples.
> >
> > With regards to the evolution of software, our datasets are derived from the 50K-C dataset (Martins et al., 2018) which a valid and compilable dataset accepted by the community. And as you ascertain that is not clear if the evolution will degrade the data quality in the future, in that case, our benchmark datasets and representations would still rely on the standard release of the 50K-C dataset. In general, evolution of datasets is a shared concern in many avenues of research, and more work in this area is needed.
>
>
>
> ****
>
>
>
> **Concern 5:** The proposed benchmark data and code are not available for replication purpose.
>
>
>
> >Clarification: We plan to release all of the prepared datasets and the code for replication after the notification. Since this will be a public benchmark, anyone interested in participating is welcome to work on the datasets and evaluate their models.
>
>
>
> ****
>
>
>
> **Concern 6:** In Table 2, the baseline result for Transformer-based method completion is missing.
>
>
>
> >Clarification: The performance for the transformer-model for the method call completion task is now available. The transformer accuracy for the method call completion task is 0.534 and is be added to the updated version of the paper.
>
> ****

---

> ### Author Response · Authors · 2020-11-13
> **Response to concerns Pt.I**
>
> We would like to thank the reviewers for their time and valuable feedback. Below are some common clarifications for concerns shared by several reviewers.
>
> -   The goal of GLUECode is to provide a benchmark that tests both local and global properties. We thus try to balance the tasks for both settings, including tasks that have been addressed in the literature locally, but could benefit from more global information.
>
> -   The GLUECode dataset is extracted from a corpus of compilable Java code. What adds a greater value to our datasets, beyond simply scraping GitHub projects, is the added parsability and compilability of projects. Such a setting allows us to run a greater number of tools, which includes a variety of static analysis tools, to procure new labels and representations for additional tasks. Cross-file information, which is useful for the global tasks, could not have been made available otherwise.
>
> -   In keeping with the spirit of a public benchmark, we confirm that we do plan to release all the datasets and relevant code publicly.
>
> -   The performance for the transformer-model for the method call completion task is now available. The transformer accuracy for the method call completion task is 0.534 and is added to the updated version of the paper. The missing reference in Section 2.2 is now resolved.
>
>
> Below, we provide a response to your concerns:
>
> ****
>
> **Concern 1:** However, it is not clear what global context is considered and how it is incorporated by the benchmark. In a ML for SE work, researchers may use various global contexts such as UML diagrams, library/API dependency, inter-procedural data/control flow, commit data, etc. It is not clear how these global context information can be satisfied by the benchmark.
>
>
>
> > Clarification: To get as much as global context information possible for a target method, one could consider the context information of the entire project. Therefore, we catalog all the methods present in a project along with their different representation types, including the raw code text representation. Some of these representations contain data/control flow information inherently; while some other global information types can be derived from the raw code. Lastly, we provide the call-graph for the project, connecting all the callers and the callees of the target method, with the cataloged methods along with their representations. Provision of such information edifice allows us to incorporate a broad global context based on call-graphs.
> >
> >Thus the goal of the benchmark is to steer research in the direction of more global contexts, but this is only a first step. With regard to additional global information such as commits or UML models, additional contexts would be interesting, but we think it is probably too challenging as a first step.
>
> ****
>
> **Concern 2:** The authors can also describe more about the unique advantages of using the proposed benchmark. Currently, they are already many public datasets released by various papers in this field (thanks to the open science policy). Also, it is easy for researchers to download a large amount of source code from open source websites (such as Github) themselves. They can also process the source code using existing static analysis tools to obtain the data they need and share the data.
>
>
>
> >Clarification: Although downloading a large set of projects from GitHub is possible, compiling those projects at scale and extracting semantic facts is a non-trivial task that none of the existing datasets perform. These semantic facts (e.g. inferred types, dependencies, call graphs, etc are an important aspect for reasoning at a more global level. Clearing this hurdle for other researchers is likely to significantly ease their work.
>
>
> ****
>
>
> **Concern 3:** Currently, GLUECode only provides a few types of source code representations. In recent years, researchers have proposed many different ways of representing source code tokens and ASTs. As an example, the following works use different AST-based source code representations (and it is not clear if the benchmark could provide necessary information to support these representations):
>
>
>
> Yao Wan, Zhou Zhao, Min Yang, Guandong Xu, Haochao Ying, Jian Wu, and Philip S. Yu. Improving automatic source code summarization via deep reinforcement learning. In ASE, pages 397–407. ACM, 2018.
>
>
>
> J. Zhang, et al., A Novel Neural Source Code Representation based on Abstract Syntax Tree, In Proc. the 41th International Conference on Software Engineering (ICSE 2019), Montreal, Canada, 2019.
>
>
> > Clarification: While we provide a single pre-processed AST representation for every sample, we think that post-processing it to transform it in another variant should be possible in general. Should there be a specific need that is not covered in our representation, we also provide the raw code of every data sample. From a reading of the papers mentioned, adapting our representation is feasible.
> ****

---

### Official Review · AnonReviewer2 · 2020-10-26
**Good objective but weak tasks and baselines**

**Rating:** 4
**Confidence:** 5

**Review:**

The objective of this paper is to present a benchmark of code understanding tasks in the spirit of GLUE benchmarks in NLP. Towards this, it designs 5 Java language tasks: NPath complexity, operator prediction, method naming, completion of method calls, and null dereference prediction. An evaluation on some common neural architectures is performed.

The first weakness of the paper is that the benchmark tasks do not fulfil the stated objective of the paper. The main argument of the paper is that many approaches in the literature focus on local (intra-procedural) prediction tasks and use either sequence representations or structured representations (e.g., ASTs, control and data flow graphs). This paper seeks to present a benchmark of tasks which requires going beyond this, by requiring global (inter-procedural) analyses and structured representations. This is a good objective and the community would certainly benefit from such a benchmark. However, among the proposed tasks, except for the null deference analysis, none of the tasks particularly require global reasoning. The NPath complexity, operator prediction, method naming and code completion (whose special case focussing on method calls in considered in this paper) are local in scope and have been solved in the literature as such.

For the null dereference prediction task, the paper uses a static analysis tool, Infer, to obtain labels. Infer is stated to return an execution path exhibiting null pointer deference. However, the paper does not give the exact number or percentage of examples from the dataset in which the paths do span multiple methods. Among all the tasks and data points, only these can be said to be truly requring global reasoning; but these details are missing and compared to the entire benchmark, this represents a small fraction. The paper conjectures that method naming and method call completion can benefit from global reasoning, but offers no evidence to that effect. I also have some concerns about the call-graph precision and ambiguity in null dereference task; these are listed in the detailed comments below.

This leads to the second weakness of the paper: the baselines. First, the baseline methods consider only sequence based models even though the paper explicitly wants to promote structured representations. They are also not tuned enough. Second, the paper does not use any global reasoning in the baselines. The paper would be more convincing if it were to show that such a global model outperforms local models. This would help to concretely claim that at least some of the benchmark tasks require global reasoning, including method naming and method call completion as conjectured by the authors. I also have other concerns above the experiments, which I list below.

Now, some detailed comments:
* The abstract says that "However, these models are commonly designed to perform well on a single task, failing to capture code’s multifaceted nature." I don't agree that just because a paper targets a single task, it fails to capture the multi-faceted nature of code. There are ample examples in the literature which take many views (e.g., ASTs, control flow, data flow, etc.) into account while solving a particular task.
* I like that the benchmarks come with pre-processed inputs in different formats. However, are the call-graphs over-approximate or under-approximate? Which call-graph construction algorithm is used? Does it construct context-sensitive call-graphs? It is important to spell out these details since different call-graph construction algorithms offer different precisions and these would impact the precision of the models build using those representations.
* There is an unresolved ref in Sec 2.2.
* The code completion task is restricted to method calls. Does this include predicting method names or their parameters also?
* The labels of the null dereference prediction task are tokens from the vocabulary (a classification problem). Such a token may occur in multiple places in the method. As stated in the paper, Infer provides the actual dereference token susceptible to null dereference. So this task should use a pointer that localizes the bug to the specific token, along the lines of "Neural Program Repair by Jointly Learning to Localize and Repair" (ICLR'19).
* The baselines are not tuned enough. There is no hyper-parameter search. The datasets vary in sizes and characteristics and would benefit from appropriate hyper-parameters.
* The paper uses a closed vocabulary of 10K. It should report on the prevelance of OOV tokens in inputs and output labels.
* The seq2seq baseline could benefit by an attention layer.
* There is no description of the task-specific layers in the Transformer baseline. The results for the completion task are not made available for review.
* The relative performance of the baselines on the NullToken task is surprisingly. The authors should explain this.
* I did not understand the argument against comparison with previous work in Sec 4.1.
* It seems that code deplication between training and test sets is not entirely ruled out. This should be fixed.

---

> ### Author Response · Authors · 2020-11-13
> **Response to concerns Pt.II**
>
> ****
> **Concern 2:** The abstract says that "However, these models are commonly designed to perform well on a single task, failing to capture code’s multifaceted nature." I don't agree with that just because a paper targets a single task, it fails to capture the multi-faceted nature of code. There are ample examples in the literature which take many views (e.g., ASTs, control flow, data flow, etc.) into account while solving a particular task.
>
> > Clarification: We precede our statement with “A multitude of machine learning models for source code have been proposed in the recent years capturing various aspects of the inherent rich structure and semantics of code” acknowledging the existence of models which take many views into account. However, these models are commonly designed to perform well only on a single task.
> >
> > By code’s multifaceted nature, indeed we mean the broader general approximation for code. Yes, even though there are many examples in the literature which take into account several views such AST with control and data flow information, which are still few and far between, they still miss out on other code properties that might be relevant to further downstream tasks. In that context, our statement merely implies that while capturing code properties for solving individual tasks many other aspects are left aside.
>
>
>
> ****
>
>
>
> **Concern 3:** The code completion task is restricted to method calls. Does this include predicting method names or their parameters also?
>
>
>
> > Clarification: At this point, the models just predict the method names, not the parameters. Predicting just the correct method name would suffice for now, hard enough as it is, because eventually when such completion models are deployed on usage-platforms such as IDEs, a correct method call prediction as top-1 prediction would be enough regardless of the number of parameters. In the future, once models are able to solve method call completion, predicting the correct parameters would be a good test to evaluate on.
>
>
>
> ****
>
>
>
> **Concern 4:** The seq2seq baseline could benefit by an attention layer.
>
>
>
> > Clarification: When it comes to baselines, there are a number of combinations we could evaluate with. However, we would encourage the community working on learning-based models of code to further improve on the baseline models which we have evaluated.
>
>
>
> ***
>
>
>
> **Concern 5:** There is no description of the task-specific layers in the Transformer baseline.
>
>
>
> > Clarification: It is a standard RoBERTa linear classification head with dropout, for the transformer model.
>
>
>
> ****
>
>
> **Concern 6:** The results for the completion task are not made available for review.
>
>
>
> > Clarification: The performance for the transformer-model for the method call completion task is now available. The transformer accuracy for the method call completion task is 0.534 and shall be added to the updated version of the paper.
>
>
>
> ****
>
>
>
> **Concern 7:** The relative performance of the baselines on the NullToken task is surprisingly. The authors should explain this.
>
>
>
> > Clarification: The simpler models such as MLP, LSTM, and CNN seem to be faring somewhat better. Subsequent evaluations on the null token prediction task showed a variance in accuracies for the simpler models and we need to conduct further diagnostic studies on them. We can report on them more comprehensively in a short time.
>
>
>
> ****
>
>
>
> **Concern 8:** I did not understand the argument against comparison with previous work in Sec 4.1.
>
>
>
> “Some of our tasks (code completion and method naming) exist in previous work. While comparing with the literature would be insightful, it is difficult, as our task formulation (and our dataset) are quite different.”
>
>
>
> > Clarification: Sorry for the misunderstanding. What we mean is that since we use both a different dataset and different evaluation metrics (for reasons mentioned in the paper), doing an apples-to-apples comparison is not feasible. Thus, while we could compare performance on our tasks with numbers published in the literature, any comparison would have to be taken with a grain of salt, which is why we refrain from doing so.
>
>
>
> ****
>
>
>
> **Concern 9:** It seems that code duplication between training and test sets is not entirely ruled out. This should be fixed.
>
>
>
> > Clarification: We have carefully checked all of our datasets and can ensure that there is no duplicated code between the training and test sets. For two of the tasks with large number of samples, we even went a step further to ensure that the datasets are project-balanced, meaning that the test set only contains samples from projects not used in the training set and there is no duplicated code even for a large number of samples.
>
> ****
>
>
> Additional clarifications regarding:
> -   percentage of examples in which the paths span multiple methods
> -   evaluation on global models
> -   call-graph construction details
>
> will be added soon.

---

> ### Author Response · Authors · 2020-11-13
> **Response to concerns Pt.I**
>
> We would like to thank the reviewers for their time and valuable feedback. Below are some common clarifications for concerns shared by several reviewers.
>
> -   The goal of GLUECode is to provide a benchmark that tests both local and global properties. We thus try to balance the tasks for both settings, including tasks that have been addressed in the literature locally, but could benefit from more global information.
>
> -   The GLUECode dataset is extracted from a corpus of compilable Java code. What adds a greater value to our datasets, beyond simply scraping GitHub projects, is the added parsability and compilability of projects. Such a setting allows us to run a greater number of tools, which includes a variety of static analysis tools, to procure new labels and representations for additional tasks. Cross-file information, which is useful for the global tasks, could not have been made available otherwise.
>
> -   In keeping with the spirit of a public benchmark, we confirm that we do plan to release all the datasets and relevant code publicly.
>
> -   The performance for the transformer-model for the method call completion task is now available. The transformer accuracy for the method call completion task is 0.534 and is added to the updated version of the paper. The missing reference in Section 2.2 is now resolved.
>
>
> Below, we provide a response to your concerns:
>
> ****
>
> **Concern 1:** Except for the null deference analysis, none of the tasks particularly require global reasoning. The NPath complexity, operator prediction, method naming and code completion are local in scope and have been solved in the literature as such. The paper conjectures that method naming and method call completion can benefit from global reasoning, but offers no evidence to that effect.
>
>
>
> > Clarification: Given that the transformer model performs quite well in comparison, for the first two tasks i.e. npath complexity and operator prediction, while struggling to score well on the other tasks provides initial evidence of needing more context information. However, we truly value your concern raised here, and it is clear that adding further baselines with global context will shed some light upon this issue.
> >
> > We would like to add that with the exception of npath complexity prediction and null token prediction, there is a good amount of related work that tackles these problems mentioned, but there is still ample room for improvement on these tasks.
>
> ****

---

### Official Review · AnonReviewer3 · 2020-10-27
**useful suite**

**Rating:** 6
**Confidence:** 5

**Review:**

### Summary ###

The paper presents GlueCode, a new benchmark suite for evaluating source code learning models. The suite includes 5 tasks, two of which are classification tasks and three are sequence generation tasks.

### Strengths ###

* A standard benchmark for evaluating source code models would be a blessing.

* The selected tasks are interesting and compelling. Particularly interesting are the tasks that require fine-grained reasoning about the control and data flow of programs. The balance between classification and generation tasks is also solid. Other design choices like focusing on the scope of a single method also seem well justified considering the common practice in this area.

### Weaknesses ###

* It is hard (impossible) to evaluate the contribution of this paper without looking at the actual code and data. The devil is in the details. On the face of it, the suggested benchmark suite seems reasonable.

* The only contribution of this paper is the benchmark suite. There is no additional novelty. This is not really a weakness, just a comment. I think that we should accept benchmark papers that help move the research area forward.


### Comments ###

* The operator prediction task seems “too easy” when only a single operator is masked. It is worth considering variations of this task when masking multiple operators.

* For the code completion task, it should be clear whether comments are part of the permitted/desired prediction context. In recent work, we are seeing increasing importance of natural language hints, and an explicit decision is required about this in the benchmark suite.

### Questions for Authors ###

* Can you please make the code and data available? All the described tasks make sense, the choice of baselines looks good.

### Minor questions and comments ###

* "Across all representations, source code entities (methods and classes) are identified via a Universally Unique Identifier (UUID), and can be linked together. ?? provides details and examples."

### A general comment about benchmarking papers ###

As a benchmark suite, this seems like a good step in the right direction, and I am happy to increase my score based on that. However, my current score is calibrated to take novelty into account when comparing to other papers.

---

> ### Author Response · Authors · 2020-11-13
> **Response to concerns**
>
> We would like to thank the reviewers for their time and valuable feedback. Below are some common clarifications for concerns shared by several reviewers.
>
> -   The goal of GLUECode is to provide a benchmark that tests both local and global properties. We thus try to balance the tasks for both settings, including tasks that have been addressed in the literature locally, but could benefit from more global information.
>
> -   The GLUECode dataset is extracted from a corpus of compilable Java code. What adds a greater value to our datasets, beyond simply scraping GitHub projects, is the added parsability and compilability of projects. Such a setting allows us to run a greater number of tools, which includes a variety of static analysis tools, to procure new labels and representations for additional tasks. Cross-file information, which is useful for the global tasks, could not have been made available otherwise.
>
> -   In keeping with the spirit of a public benchmark, we confirm that we do plan to release all the datasets and relevant code publicly.
>
> -   The performance for the transformer-model for the method call completion task is now available. The transformer accuracy for the method call completion task is 0.534 and is added to the updated version of the paper. The missing reference in Section 2.2 is now resolved.
>
>
> Below, we provide a response to your concerns:
>
> ****
>
> **Concern 1:** The operator prediction task seems “too easy” when only a single operator is masked. It is worth considering variations of this task when masking multiple operators.
>
> >Clarification: In case you refer to the transformer’s performance specifically, we think this could be due to the masked language modelling pretraining which might be an appropriate pre-training task for this specific case. In case you are thinking of something else, it would help if you could clarify a bit more. Regardless, constructing a multi-masked operator prediction task would make it much harder indeed.
>
> ****
>
> **Concern 2:** For the code completion task, it should be clear whether comments are part of the permitted/desired prediction. In recent work, we are seeing increasing importance of natural language hints, and an explicit decision is required about this in the benchmark suite.
>
> >Clarification: Comments are available in the raw code representation for every data sample, it is up to the end-user to decide whether they’d like to use them in their chosen representations for their model predictions. Thus, the usage of comments for additional context is permissible for our benchmark datasets.
>
> ****
>
> **Concern 3:** Can you please make the code and data available?
>
>
> > Clarification: We plan to release all of the prepared dataset and the code for replication after the notification. Since this will be a public benchmark, anyone interested in participating is welcome to work on the dataset and evaluate their models, and use our code.
>
> ****

---

### Official Review · AnonReviewer1 · 2020-10-28
**GLUECode: A Benchmark for Source Code Machine Learning Models**

**Rating:** 4
**Confidence:** 3

**Review:**

Reasons for score:
A benchmark for evaluating source code ML models will help to accelerate the progress in the right direction. However, the analysis to support the dataset and the proposed tasks as a benchmark does not address some critical concerns (please see weakness section below).

Summary:
The paper presents a dataset of source code that allows experimenting with different representations and proposes five tasks to evaluate local and global reasoning capabilities of a source code machine learning model.

Strength:
1. GLUECode provides a labeled dataset with different representations by compiling ~5300 Java projects extracted from Github. This could be useful for future research in ML modeling for source code.
2. Evaluation results of five baseline models on five proposed benchmark tasks demonstrate varying performances over different tasks.

Weakness
1. Although overall construction of the dataset could be useful for the community, sufficient evidence is not provided to establish the utility of the dataset compared to other existing datasets. An arbitrary minimum number of 50 files in a project is selected as a filtering method without presenting any supporting analysis. “NullToken” task is presented as the task that benefits most from global reasoning, which is the primary contribution of this paper. However, the dataset size for NullToken task is small, which significantly reduces the usefulness of the task in evaluating the models.
2. The evaluation results of the baseline models are not well-explained. According to Table-2 the “Completion” task requires increased non-structural and global reasoning compared to the “Naming” task. However, all the baseline models are showing poor performance in the “Naming” task compared to the “Completion” task.
3. The tasks that require global reasoning are mostly generation tasks in the benchmark. Therefore, the evaluation metrics could be less representative of global-reasoning performance of classification models.

Questions to author:
Please address and clarify the cons above.

---

> ### Author Response · Authors · 2020-11-13
> **Response to concerns Pt.II**
>
> ****
> **Concern 4:** The evaluation results of the baseline models are not well-explained.
> > Clarification: The performance of the transformer-model for the method call completion task is now added to the updated revision. Subsequently we explain the results here.
> >
> > Overall, we see that the Transformer exhibits higher performance on the first four tasks (NPath prediction, Operator prediction, Method naming), and has reasonably acceptable performance only on the first two tasks (Npath prediction and Operator prediction), which are the most local ones.
> >
> > For the tasks which have some globalness aspect to it, the transformers have an average accuracy of ~40% with highest score being barely above the fifty percent threshold for the method call completion task. Even in the local tasks, where the transformers score well, there is still a margin for improvement of more than 20%.
>
> >**NPTH:** For npath complexity prediction task, the transformer model is the best performing model, with ~75% accuracy, followed by the sequence to sequence model with ~54% accuracy. The sequence to sequence model is able to encode the complexity from the input code into a single embedding which could then be rendered correctly as output. The transformer model using multi-head attention performs reasonably better. Further, the simple MLP shows the least-favorable performance for this task, with BiLSTM and CNN models doing marginally better.
>
> >**OPER:** For the operator prediction task, the transformer model performs the best, while the CNN seems to be worst-performing model for the given dataset. CNN’s are good at extracting position-invariant features, but since operator prediction needs important sequential information, it fares poorly in comparison.
> The BiLSTM model does comparatively good, as they are designed to make use of sequential data. Since, RNNs usually are good at predicting what comes next in a sequence, the BiLSTM model is second only to the transformer model. The sequence to sequence model does barely better than the baseline MLP, since sequence to sequence models encode the masked code to generate a single embedding which can effectively summarize certain properties of the code. And since operators do not represent a code property per se that can be translated into an output, at best the models could establish only simple associations between the input and output.
>
> >**NAME:** For methodnaming, the transformer model shows the best performance, followed by the sequence to sequence model, and then the BiLSTM model. For method naming, performance is much lower; it is also lower than in similar naming tasks, but having evaluated with different metrics, it shows that our choices yield a more challenging task.
>
> >**COMP:** Once again, for the method call completion task, the transformer model shows the best performance, followed by the sequence to sequence model, and then the BilSTM model.
> It is important to note here that unlike method naming, completion task has many labels (method api calls) which belong to the Java standard library, such as println(), toString() etc. which are commonly used, and which are easier to predict for DL models (Hellendoorn et al.,2019a). About 20% of the dataset consist of standard library method calls. This might explain why the models perform better in comparison solely against the method naming task.
>
> >**NTKN:** Finally, we observe that on the Null Token prediction task performance is very low, even the Transformer model is not faring well here, especially considering that a naive baseline would score 20%, and models are barely better than this, indicating opportunity for further progress. The simpler models such as MLP, LSTM, and CNN seem to be faring somewhat better. Subsequent evaluations on the null token prediction task showed a variance in accuracies for the simpler models and we need to conduct further diagnostic studies on them.
> ****
> **Concern 5:** According to Table-2 the “Completion” task requires increased non-structural and global reasoning compared to the “Naming” task. However, all the baseline models are showing poor performance in the “Naming” task compared to the “Completion” task.
> > Clarification: As mentioned, unlike naming, the code completion task has many labels (method calls) which belong to the Java standard library, such as println(), toString() etc. which are commonly used, and can be learned easily.
> >
> >About 20% of the dataset consist of standard library method calls. This might seem to help the models to perform better when comparing solely against the naming task. A brief explanation was given in Section 3.1 of the paper.
> ****
> **Concern 6:** The tasks that require global reasoning are mostly generation tasks in the benchmark. Therefore, the evaluation metrics could be less representative of global-reasoning performance of classification models.
>
> > If you could clarify this statement a bit, we would be able to respond to your concern better.
> ****

---

> > ### Comment · AnonReviewer1 · 2020-11-16
> > **Clarification about missing classification task that requires global reasoning.**
> >
> > There is no classification task in the benchmark that relies on the global properties. Currently, the evaluation of this property is represented by some generation tasks. As one of the motivation for the proposed benchmark is to introduce tasks that require understanding of the global properties, a classification task with global scope would make the benchmark more complete.

---

> > > ### Author Response · Authors · 2020-11-18
> > > **RE: Clarification about missing classification task that requires global reasoning**
> > >
> > > ****
> > > **Concern 6:**  The tasks that require global reasoning are mostly generation tasks in the benchmark. Therefore, the evaluation metrics could be less representative of global-reasoning performance of classification models.
> > >
> > > There is no classification task in the benchmark that relies on the global properties. Currently, the evaluation of this property is represented by some generation tasks. As one of the motivation for the proposed benchmark is to introduce tasks that require an understanding of the global properties, a classification task with global scope would make the benchmark more complete.
> > >
> > > > We can include other tasks in an extended version of the benchmark with classification as the modus operandi for some global tasks; however, since our benchmark is more interested in how models approximate the sample data features, adding or adapting tasks with a classification head serves only as a peripheral need.
> > > >
> > > > We would like to better understand your concern about generation tasks hindering global-reasoning performance compared to classification tasks. Could you please explain why you believe this is the case?
> > >
> > > ****

---

> > > > ### Comment · AnonReviewer1 · 2020-11-24
> > > > **Concern about missing classification tasks is reduced.**
> > > >
> > > > Thanks for your response. I agree that adding tasks with a classification head may not always result in any additional gain in understanding models performance. Also, generation tasks may not impact global-reasoning performance differently than the classification task. I was considering scenarios where models designed for generation tasks could have different performance requirement than the models designed for classification tasks. Adding classification task would provide additional results from the benchmark for comparing different models designed for classification tasks.

---

> ### Author Response · Authors · 2020-11-13
> **Response to concerns Pt.I**
>
> We would like to thank the reviewers for their time and valuable feedback. Below are some common clarifications for concerns shared by several reviewers.
>
> -   The goal of GLUECode is to provide a benchmark that tests both local and global properties. We thus try to balance the tasks for both settings, including tasks that have been addressed in the literature locally, but could benefit from more global information.
>
> -   The GLUECode dataset is extracted from a corpus of compilable Java code. What adds a greater value to our datasets, beyond simply scraping GitHub projects, is the added parsability and compilability of projects. Such a setting allows us to run a greater number of tools, which includes a variety of static analysis tools, to procure new labels and representations for additional tasks. Cross-file information, which is useful for the global tasks, could not have been made available otherwise.
>
> -   In keeping with the spirit of a public benchmark, we confirm that we do plan to release all the datasets and relevant code publicly.
>
> -   The performance for the transformer-model for the method call completion task is now available. The transformer accuracy for the method call completion task is 0.534 and is added to the updated version of the paper. The missing reference in Section 2.2 is now resolved.
>
> Below, we provide a response to your concerns:
>
> ****
> **Concern 1:** Although overall construction of the dataset could be useful for the community, sufficient evidence is not provided to establish the utility of the dataset compared to other existing datasets.
>
>
>
> > Clarification: The utility of our dataset and tasks is twofold: first, GLUECode is the only benchmark that provides tasks that both require local and global reasoning. Second, it provides the building blocks (including several base code representations) for researchers to experiment with the models that can solve the tasks. Additionally, GLUECode’s dataset is extracted from a corpus of compilable Java code. And beyond simply scraping GitHub projects, our dataset allows compilability of projects. Such a setting allows us to run a greater number of tools, which includes static analysis tools, to procure new labels and representations for additional tasks. Cross-file information (e.g. useful for completion and null dereference tasks) could not be made available otherwise.
>
>
> ****
> **Concern 2:** An arbitrary minimum number of 50 files in a project is selected as a filtering method without presenting any supporting analysis.
>
> > Clarification: We used 50 as a heuristic for detecting small, possibly immature or toy projects. By filtering projects with more than 50 files (=classes in Java), we get a sufficient number of projects that have rich structure.
>
> ****
>
> **Concern 3:** “NullToken” task is presented as the task that benefits most from global reasoning, which is the primary contribution of this paper. However, the dataset size for NullToken task is small, which significantly reduces the usefulness of the task in evaluating the models.
>
> >Clarification: We see the small number of samples in a different light: we see it as an incentive to promote more sample-efficient models, whether by leveraging pre-training or additional structure in the data. We also note that gathering even this limited amount of data is not trivial, as it involves running a costly static analysis on 5000+ compilable software projects that are large enough.
>
> ****

---

### Author Response · Authors · 2020-11-13
**General clarifications**

We would like to thank the reviewers for their time and valuable feedback.
Below are some common clarifications for concerns shared by several reviewers.

- The goal of GLUECode is to provide a benchmark that tests both local and global properties. We thus try to balance the tasks for both settings, including tasks that have been addressed in the literature locally, but could benefit from more global information.

- The GLUECode dataset is extracted from a corpus of compilable Java code. What adds a greater value to our datasets, beyond simply scraping GitHub projects, is the added parsability and compilability of projects. Such a setting allows us to run a greater number of tools, which includes a variety of static analysis tools, to procure new labels and representations for additional tasks. Cross-file information, which is useful for the global tasks, could not have been made available otherwise.

- In keeping with the spirit of a public benchmark, we confirm that we do plan to release all the datasets and relevant code publicly.

- The performance for the transformer-model for the method call completion task is now available. The transformer accuracy for the method call completion task is 0.534 and is added to the updated version of the paper. The missing reference in Section 2.2 is now resolved.

---

### Decision · Program_Chairs · 2021-01-07
**Final Decision**

**Decision:**

Reject

**Comment:**

This paper proposes a new source code modeling benchmark, with the unique twist being that we not only have code source text, but we also have build information, which allows extracting richer information to construct labels from. This enables, for example, a null pointer prediction task with labels coming from an inter-procedural static analysis tool. AC and reviewers agree that this is a valuable framing for a benchmark suite. Unfortunately, it’s not clear that the benchmark in its current form delivers on the promise of the framing. Much of the interest and novelty is limited to just the one NullToken task, and reviewers raise a number of concerns including dataset size and whether the task truly measures the inter-procedural reasoning that it sets out to measure. AnonReviewer2 raised some good questions here that the authors promised to address in a forthcoming comment, but that didn’t come before the discussion deadline. I’d encourage the authors to use the reviewer suggestions to more strongly establish that these tasks measure what they set out to measure, and also to consider adding other tasks that measure whether our ML models are capable of deeper / longer-range reasoning. In total, there is a lot of potential here, but the work needs another iteration before it’s ready for publication.